# Circulating Extracellular Vesicle Proteins and MicroRNA Profiles in Subcortical and Cortical-Subcortical Ischaemic Stroke

**DOI:** 10.3390/biomedicines9070786

**Published:** 2021-07-07

**Authors:** Laura Otero-Ortega, Elisa Alonso-López, María Pérez-Mato, Fernando Laso-García, Mari Carmen Gómez-de Frutos, Luke Diekhorst, María Laura García-Bermejo, Elisa Conde-Moreno, Blanca Fuentes, María Alonso de Leciñana, Susana B. Bravo, Exuperio Díez-Tejedor, María Gutiérrez-Fernández

**Affiliations:** 1Neurological Sciences and Cerebrovascular Research Laboratory, Department of Neurology and Stroke Center, La Paz University Hospital, Neuroscience Area of IdiPAZ Health Research Institute, Universidad Autónoma de Madrid, Paseo de la Castellana 261, 28046 Madrid, Spain; oteroortega.l@gmail.com (L.O.-O.); elisaalonso164@hotmail.com (E.A.-L.); mery19832005@yahoo.es (M.P.-M.); fernilaso.9@gmail.com (F.L.-G.); mcarmen.gomezf@gmail.com (M.C.G.-d.F.); luke.diekhorst@gmail.com (L.D.); blanca.fuentes@salud.madrid.org (B.F.); malecinanacases@salud.madrid.org (M.A.d.L.); exuperio.diez@salud.madrid.org (E.D.-T.); 2Biomarkers and Therapeutic Targets Unit, Instituto Ramón y Cajal de Investigación Sanitaria (IRYCIS), 28034 Madrid, Spain; garciabermejo@gmail.com (M.L.G.-B.); elisa.condem@gmail.com (E.C.-M.); 3Proteomic Unit, Instituto de Investigaciones Sanitarias-IDIS, Complejo Hospitalario Universitario de Santiago de Compostela (CHUS), 15706 Santiago de Compostela, Spain; sbbravo@gmail.com

**Keywords:** exosomes, extracellular vesicles, ischaemic stroke, miRNA, proteomic analysis

## Abstract

In order to investigate the role of circulating extracellular vesicles (EVs), proteins, and microRNAs as damage and repair markers in ischaemic stroke depending on its topography, subcortical (SC), and cortical-subcortical (CSC) involvement, we quantified the total amount of EVs using an enzyme-linked immunosorbent assay technique and analysed their global protein content using proteomics. We also employed a polymerase chain reaction to evaluate the circulating microRNA profile. The study included 81 patients with ischaemic stroke (26 SC and 55 CSC) and 22 healthy controls (HCs). No differences were found in circulating EV levels between the SC, CSC, and HC groups. We detected the specific expression of C1QA and Casp14 in the EVs of patients with CSC ischaemic stroke and the specific expression of ANXA2 in the EVs of patients with SC involvement. Patients with CSC ischaemic stroke showed a lower expression of miR-15a, miR-424, miR-100, and miR-339 compared with those with SC ischaemic stroke, and the levels of miR-339, miR-100, miR-199a, miR-369a, miR-424, and miR-15a were lower than those of the HCs. Circulating EV proteins and microRNAs from patients with CSC ischaemic stroke could be considered markers of neurite outgrowth, neurogenesis, inflammation process, and atherosclerosis. On the other hand, EV proteins and microRNAs from patients with SC ischaemic stroke might be markers of an anti-inflammatory process and blood–brain barrier disruption reduction.

## 1. Introduction

Extracellular vesicles (EVs) are membrane particles secreted by most cells that constitute major vehicles for intercellular communication. In response to a stroke, EVs are released into the blood from brain cells and other organs [1,2,3,4,5]. These EVs can carry proteins and miRNAs that reveal important information regarding damage, protection, and the repair process after brain ischaemia, as well as stroke outcomes [6,7,8,9,10].

The mechanisms underlying the cerebral ischaemic cascade are highly complex and can differ depending on the lesion topography [11]. Most research studies on stroke have focused on the cortical neuronal ischaemic cascade but have ignored the subcortical damage with the involvement of white matter fibre and oligodendrocytes. Studying the subtypes of stroke that affect different locations, such as subcortical (SC) and cortical-subcortical (CSC) ischaemic stroke, could provide information on the involved damage and repair processes, depending on the affected cell type [12].

The aim of the present study was therefore to analyse whether EV levels, proteins, and microRNAs could be modified based on the differing lesion topography in patients with SC and CSC ischaemic stroke. The results would provide relevant information for routine clinical practice to develop novel therapeutic strategies according to the affected tissue area. 

## 2. Materials and Methods 

### 2.1. Study Design

This is a sub-study of a previously published article [4]. This was a clinical, prospective, and observational study that recruited patients with ischaemic stroke in the first 24 h after onset from among those admitted to the Stroke Unit of the Department of Neurology of La Paz University Hospital in Madrid between February 2017 and June 2019. The inclusion criteria were an age older than 18 years and a pre-stroke mRS score ≤1 (Figure 1A). Patients who presented with a strictly SC cerebral infarction that clinically caused a lacunar syndrome (pure sensory syndrome, pure motor syndrome, sensory-motor syndrome, dysarthria-clumsy hand, or ataxic hemiparesis) were included. Patients who presented with cerebral infarction with CSC lesion in the territory of the middle cerebral artery and/or anterior cerebral artery were also included. The exclusion criteria were a previous history of ischaemic stroke, dementia, transient ischaemic attack, brain tumour, cerebral haemorrhage, drug or alcohol dependence, any clinical condition that precluded diagnosis, and follow-up and participation in a clinical trial.

The study included HCs who voluntarily agreed to participate by signing the informed consent. 

Patients underwent magnetic resonance imaging (MRI) or computed tomography (CT) of the brain in order to confirm the diagnosis and type of ischaemic stroke. We recorded the participants’ demographics, risk factors, type of stroke, baseline stroke severity (using the NIHSS), and treatment undergone (intravenous thrombolysis alone, mechanical thrombectomy alone, or both). We assessed the 3-month outcomes using the NIHSS score and mRS. 

All participants or their proxies (in cases of aphasia, confusion, or reduced level of consciousness) provided their informed consent after a detailed explanation of the nature and purpose of this study, which was approved by the Clinical Research Ethics Committee of La Paz University Hospital (PI-2562).

Infarct volume was analysed at 1 week in a subgroup of 39 patients who needed an MRI for medical reasons (23 CSC and 16 SC). The final volume of the cerebral infarct was analysed using diffusion-weighted magnetic resonance imaging (DWI) and fluid-attenuated inversion recovery (FLAIR) sequences. The infarct area was manually delimited in all axial slices acquired in DWI and FLAIR sequences using the MRIcron program (Chris Rorden, CA, USA). The total volume in cubic centimetres was calculated from the number of voxels and the image dimension. 

To determine any significant differences in baseline vascular disease burden, we analysed white matter lesions of presumed vascular origin and previous silent brain infarcts using the Fazekas score as shown on CT scans [13].

### 2.2. Blood Extraction, Extracellular Vesicle Isolation and Characterisation

Blood samples were collected within 24 h following stroke onset. The tubes were centrifuged at 3000× *g* for 15 min at 4 °C, and the serum samples were stored at −80 °C until analysis.

We employed ExoQuick Ultra EV precipitation solution (System Biosciences, Palo Alto, CA, USA) to precipitate the serum EVs, as previously described [14]. This method isolates the EVs with almost no contamination by albumins and immunoglobulins, as previously reported [4]. 

To characterise the EVs, we examined their mean size and morphology by viewing them through a nanoparticle tracking analysis using a NanoSight LN10 instrument (Malvern Instruments Ltd., Malvern, UK) (Figure 1B), which was performed in triplicate. We employed a scanning electron microscope to visualise the obtained EVs and measure their sizes (JEOL JEM1010) (Figure 1C), a procedure also performed in triplicate. Lastly, we performed Western blotting using markers anti-CD63 (1:250, Abcam, Cambridge, UK), anti-Alix (1:250, Cell Signal, Danvers, MA, USA), and anti-CD81 (1:250, Abcam, Cambridge, UK) antibodies, followed by goat anti-mouse and anti-rabbit Alexa Fluor 488 antibodies (1:750, Invitrogen, Waltham, MA, USA). We used albumin as the negative control. The procedure was performed in triplicate. The blots were visualised using chemiluminescence and an Uvitec–Cambridge imaging system (Figure 1D,E). 

### 2.3. Quantification of Circulating Extracellular Vesicles

We analysed the circulating EV levels in the SC, CSC, and HC groups and compared them. EVs were quantified in 50 µL of serum from each patient using an ELISA kit EV EXOEL-CD63A-1 enzyme-linked immunosorbent assay kit (System Biosciences, Palo Alto, CA, USA) based on the presence of the CD63 marker. 

### 2.4. Proteomic Analysis of Circulating Extracellular Vesicles

In order to analyse the proteome content of the EVs in each study group, we pooled the blood samples from the SC, CSC, and HC groups and compared the intergroup findings. After isolating them using the ExoQuick ULTRA protocol, the EVs were lysed in bulk, and proteins were processed through sodium dodecyl sulphate–polyacrylamide gel electrophoresis (SDS-PAGE) using a 10% Bis Tris NuPage mini-gel (Invitrogen, Waltham, MA, USA). The migration windows (1 cm gel lane) were excised and processed by in-gel digestion with trypsin using a ProGest robot (DigiLab, Hopkinton, MA, USA). For mass spectrometry, the gel digest was analysed by nano liquid chromatography with tandem mass spectrometry using a Waters nanoACQUITY HPLC system interfaced to a ThermoFisher Scientific Q Exactive mass spectrometer, which was operated in data-dependent mode, with the Orbitrap operating at 70,000 full width half maximum and 17,500 full width half maximum for mass spectrometry and tandem mass spectrometry, respectively. Data files were parsed into Scaffold (Proteome Software, Portland, OR, USA) for validation, filtering, and to create a nonredundant list per sample. 

### 2.5. Validation of Selected Proteins

We also validated the results of this proteomic study, analysing the proteome content of 10 isolated patients per group who were chosen by age and sex matching and for having similar lesions. After isolating the EVs using the ExoQuick ULTRA protocol, we lysed the EVs in bulk and extracted the proteins from a 10% SDS-PAGE gel, as previously described [15,16]. 

### 2.6. Proteomic Analysis of Neuron-Derived Extracellular Vesicles

Most studies have isolated the neuron-derived EVs directly from brain tissue [17]. Taking into account the translational nature of the present study, we decided to use a less invasive method for isolating circulating neuron-derived EVs from the serum. Isolating pure circulating brain-derived EVs, without contaminants, is extremely challenging. For these reasons, our EV preparations are enriched in brain-derived EVs rather than pure samples. We transferred 500 mL of serum to an EV isolation column (SmartSEC column), incubated it for 30 min at room temperature, and washed them by centrifuging at 500× *g* for 1 min with SmartSEC column buffer. The L1CAM antibody was then added directly to the pooled isolated EVs. We then transferred 350 µL of EV isolation beads to a neuron-EV isolation column in a 2 mL Eppendorf tube. The L1CAM-isolated EVs were also loaded into the neuron-EV isolation column and centrifuged. We added 300 µL of elution buffer and centrifuged it to obtain the fraction of purified neuron-derived EVs. The sample was then neutralised by adding 1/10 of the volume of the neutralisation buffer. We performed mass spectrometry of the neuron-derived EVs, as described above. 

### 2.7. RNA Extraction and Reverse Transcription Quantitative Polymerase Chain Reaction

We analysed the miRNAs in serum (given that most miRNAs are concentrated in EVs) [18] and compared them between the SC, CSC, and HC groups. We employed a synthetic RNA (spike-in) as a technical control for extraction homogeneity by further spike-in amplification. We performed the isolation of total RNA enriched in miRNAs using the RNeasy Mini Kit (Qiagen, Germantown, MD, USA) and employed external RNA (cell-miR-39) as a control of cDNA synthesis efficiency. For the cDNA synthesis, we employed the Universal RT miRNA PCR System (Qiagen, Germantown, MD, USA).

### 2.8. miRNA Array Profiling

We selected 4 patients per group for the quantitative polymerase chain reaction (qPCR) array screening as previously described [19]. These 4 patients per group were chosen by matching age and sex and having similar lesions. This quantitative PCR array screened 752 miRNAs using the miRCURY LNA miRNA miRNome PCR Panels I + II (YAHS-312 YG-8, Qiagen, Germantown, MD, USA). The array data were normalised using the average expression of all miRNAs exhibiting cycle threshold (Ct) values ≤ 34. We analysed the data with GenEx v.6 software (Göteborg, Sweden).

### 2.9. Validation of Selected miRNAs

We validated the miRNA results of the PCR array by reverse transcription (RT)-qPCR in 20 patients per group. RNA degradation was checked in all samples. PCR detection was performed using SYBR Green and specific locked nucleic acid probes for each selected miRNA (Qiagen, Germantown, MD, USA). We performed all reactions in triplicate using a Light Cycler 480 instrument (Roche, Basel, Switzerland) and calculated the Ct values using a second derivative method (Light Cycler 480 Software 1.5, Roche, Germantown, MD, USA). miRNA expression values are presented as ΔCT. The mean miR-103a-3p Ct and miR-30c-5p Ct were employed as the normaliser Ct. 

### 2.10. Association between miRNAs and Outcomes 

We measured the ability of miRNA levels to predict improvement in NIHSS scores from baseline. The primary endpoint, improvement from baseline NIHSS scores, was considered as >50% of the percentage change in NIHSS score at 3 months using the following formula: 1-(NIHSS at 3 months/baseline NIHSS) × 100 [20,21]. The secondary endpoint analysis was based on the mRS score at 3 months, and we considered a score <2 (functional independence) as a good outcome. 

### 2.11. Bioinformatic Analysis

Specific proteins from each study group were identified and described. For the gene ontology analysis, we employed Scaffold software [22] to classify the genes into distinct categories of molecular functions and pathways. Proteins were classified into families and subfamilies of shared function, which were then categorised by process and function. We employed the Search Tool for the Retrieval of Interacting Genes (STRING) [23] to study protein interactions (https://string-db.org) (last accessed on 10 March 2021). We also analysed the abundant proteins in each group as well as analysing the miRNA predicted target using MirTarget (http://www.mirdb.org) (last accessed on 10 March 2021) [24], which analyses thousands of miRNA–target interactions from high-throughput sequencing experiments. 

### 2.12. Statistical Analysis

Data were tested for normality using the Kolmogorov–Smirnov test for groups with more than 30 degrees of freedom and the Shapiro–Wilk test for groups with less than 30 degrees of freedom. We compared the study groups’ demographics, clinical data, and circulating levels of EVs and miRNAs, comparing the continuous variables using the Kruskal–Wallis test with a *post hoc* Mann–Whitney U test, as well as Fisher’s exact test for categorical variables. The correlation between circulating EV levels and final infarct volume was analysed by Spearman correlation test. Circulating EV levels (expressed as mean ± SD) and the miRNA content were compared using an analysis of variance with *post hoc* Bonferroni correction for multiple comparisons. We assessed the possible relationship between miRNAs and the main endpoint (i.e., 50% improvement from baseline NIHSS score at 3 months) using logistic regression. The model was adjusted for age and the administration of reperfusion therapies. For those mRNAs that were found to be related to outcome, we performed an ROC analysis and chose the cut-off point corresponding to the maximum sensitivity and specificity values to identify the miRNA level in order to predict a >50% improvement from baseline NIHSS score at 3 months. The possible relationship between miRNAs and the secondary endpoint (mRS < 2, i.e., functional independence) was assessed by logistic regression. For the proteomic analysis, we employed an unsupervised multivariate statistical analysis using a principal component analysis to compare the data across the samples, using scaling. The average MS peak area of each protein was derived from the replicates of the SWATH-MS of each sample, followed by Student’s *t*-test analysis using MarkerView software (ABSciex LLC, Framingham, MA, USA) to compare the samples based on the averaged area sums of all transitions derived for each protein. Proteins (*p*-value < 0.05) with a 1.5-fold increase or decrease were selected. We employed the Statistical Package for the Social Sciences (SPSS, version 19, Armonk, NY, USA: IBM Corp.) for the analysis.

## 3. Results

This is a sub-study of a previously published study [4] where we enrolled 81 patients with stroke (55 CSC, 26 SC) and 22 healthy controls (HCs) (Figure 1A). Table 1 lists their demographic and clinical characteristics. The HC values have been previously published [4]. 

### 3.1. Association between the Quantity of Circulating Extracellular Vesicles of Healthy Controls and Patients with Either Subcortical or Cortical-Subcortical Stroke

EV levels showed no significant differences between the SC (2.67 × 10^9^ [±2.91 × 10^9^] EVs/mL), CSC (3.27 × 10^9^ [±3.3 × 10^9^] EVs/mL), and HC (2.30 × 10^9^ [±2.70 × 10^9^] EVs/mL) groups at 24 h (*p* = 0.10) (Figure 1E) or between the patients with stroke (3.06 × 10^9^ [±3.16 × 10^9^] EVs/mL) and the HCs (2.30 × 10^9^ [±2.70 × 10^9^] EVs/mL) (*p* = 0.26).

### 3.2. Correlation between the Quantity of Circulating Extracellular Vesicles and Final Lesion Volume

We did not find a significant correlation (*p* = 0.406) between the EV levels at 24 h and the final infarct volume at 1 week (R = 0.012) (Figure 1F). 

### 3.3. Proteomic Analysis

Appendix A lists the total proteins detected in the circulating EVs. In total, 191 proteins were detected in the EV samples with high confidence. Among these 191 proteins, 108 were found in EV samples from all groups (Figure 2A), and 8 were isolated only in the HCs. Seven specific proteins were found only in the EVs of the CSC group, and 24 proteins were isolated only in the SC group (Figure 2B). Figure 2C presents the more abundant proteins and the less abundant proteins. All of the abundant proteins from the HCs showed physical and functional interactions. All of the most abundant proteins and the specific C1q A chain protein (C1QA) interacted in the CSC group. The specific annexin A2 protein (ANXA2) and all of the most abundant proteins showed interactions in the SC group (Figure 2D). 

### 3.4. Validation of the Identified Proteins

The complement proteins C1QA and caspase 14 (Casp14) were confirmed as specific proteins of the CSC group with validated proteomic results in 10 patients per group. Moreover, ANXA2 was found in the SC group and not in the CSC or HC groups (Figure 2B). Apolipoprotein B (APOB), complement C3, alpha-2-macroglobulin (A2M), C4b-binding protein alpha chain precursor (C4BPA), and fibronectin 1 (FN1) were validated as abundant proteins in HCs. APOB, C3, A2M, Von Willebrand factor (VWF), and FN1 were validated as abundant proteins in the SC group. APOB, C3, A2M, C4BPA, and VWF were validated as abundant proteins in CSC. Alpha-1-acid glycoprotein 1 (ORM1), serpin B12, cathepsin D (CTSD), fibrinogen A (FGA), and fibrinogen G (FGG) were validated as low proteins in HCs. S100 calcium-binding protein A9 (S100A9) and selenoprotein P were validated as low proteins in SC, and myosin heavy chain 9 (MYH9), S100 calcium-binding protein A7 (S100A7), immunoglobulin lambda variable 3-10 (IGLV3-10), and CTSD were validated as low proteins in CSC. Furthermore, the levels of the proteins fibrinogen C (FGC), C4BPA, VWF, and FN1 were validated as higher in the CSC group than in the HCs (*p* = 0.014, *p* = 0.003, *p* = 0.002, and *p* = 0.043, respectively). Moreover, ORM1 levels were validated as higher in the SC group than in the HCs (*p* = 0.005) (Figure 2C).

### 3.5. Proteomic Analysis of Neuron-Derived Extracellular Vesicles

A total of 1279 proteins were found inside the L1 cell adhesion molecule (L1CAM) EVs. Appendix A lists the total proteins detected from the neuron-derived EVs. The specific proteins of the CSC (C1QA and Casp14) and SC (ANXA2) groups were also explicitly identified within the neuron-derived L1CAM. Among the most abundant proteins, we also found APOB, C3, FN1, VWF, and FGG in the neuron-derived L1CAM.

### 3.6. Gene Ontology Enrichment

For the classification of biological processes, the top five enrichment terms were: response to stimulus, cellular processes, biological regulation, metabolic processes, and immune system processes (Figure 3A). The EVs of the SC group have two more pathways than those of the CSC group and HCs. These pathways are the dopamine receptor-mediated signalling pathway and the nicotine pharmacodynamics pathway, both of which are mediated by the filamin-A protein. In addition, the FAS signalling pathway was not found in the CSC group (Figure 3B).

### 3.7. miRNA Analysis 

We observed significantly lower levels of miR-369_5p (*p* = 0.022), miR-199a_3p (*p* = 0.009), miR-15a_5p (*p* = 0.041), miR-424_5p (*p* = 0.001), miR-100_5p (*p* = 0.012), and miR-339_5p (*p* = 0.008) in the CSC group compared with the HCs and observed higher miR-29_3p levels. Compared with the HCs, the SC group had significantly higher miR-29b_3p levels (*p* = 0.045). Moreover, we observed significantly lower levels of miR-15a_5p (*p* = 0.001), miR-424_5p (*p* = 0.002), miR-100_5p (*p* = 0.001), and miR-339_5p (*p* = 0.002) in the CSC groups compared with the SC group (Figure 4A).

### 3.8. Bioinformatics Analysis of Predicted Targets

We analysed miRNA predicted targets using a bioinformatic analysis. Our study determined that miR-424 levels were significantly lower in the CSC group than in the HCs (*p* = 0.001) and in the SC group (*p* = 0.002). VWF was the predicted target of this miRNA, and the levels of this protein were significantly higher in the CSC group than in the HCs (*p* = 0.002) and in the SC group (*p* = 0.004). FN1 was also the predicted target of miR-424 and was significantly higher in the CSC group than in the HCs (*p* = 0.043). Moreover, miR-15a levels were significantly lower in the CSC group than in the SC group (*p* = 0.001) and in the HCs (*p* = 0.041). VWF, A2M, and FN1 were the predicted targets of this miRNA; the levels of VWF and A2M were higher in the CSC group than in the SC group (*p* = 0.004 and *p* = 0.025, respectively), and the levels of VWF and FN1 were higher in the CSC group than in the HCs (*p* = 0.002 and *p* = 0.043, respectively). Lastly, the miR-199 levels were significantly lower in the CSC group than in the HCs (*p* = 0.009), and the levels of its predicted target FN1 were higher in the CSC group than in the HCs (*p* = 0.043) (Figure 4B) 

### 3.9. miRNA as a Prognostic Biomarker in Ischaemic Stroke

Of all the miRNAs, only miR-100-5p showed a relationship with improved scores on the National Institutes of Health Stroke Scale (NIHSS). A receiver operating characteristic (ROC) curve determined that miRNA-100-5p levels below a cut-off of 5.47 ΔCT predicted a greater than 50% improvement in NIHSS scores at 3 months (area under the curve 0.887, sensitivity 80%, specificity 79.2%, and *p* = 0.007) (Figure 4C). No miRNA showed a significant relationship with functional outcomes at 3 months.

## 4. Discussion 

In this study, we determined that although EV levels showed no differences between the SC, CSC, and HC groups, there were differences among these groups in the protein and miRNA content of circulating EVs. Moreover, we detected the specific expression of two proteins (C1QA and Casp14) in the EVs of the CSC group and the specific expression of ANXA2 in the EVs of the SC group. We determined that APOB, C3, A2M, C4BPA, and FN1 were abundant proteins in the HCs, while APOB, C3, A2M, VWF, and FN1 were abundant in the SC group, and APOB, C3, A2M, C4BPA, and VWF were abundant in the CSC group. Furthermore, the levels of the proteins FGG, C4BPA, VWF, and FN1 were higher in the CSC group than in the HCs, and the ORM1 levels were higher in the SC group compared with the HCs. The CSC group showed a lower expression of miR-15a, miR-424, miR-100, and miR-339 compared with the SC group, and the levels of miR-339, miR-100, miR-199a, miR-369a, miR-424, and miR-15a were lower than those of the HCs. The bioinformatic analysis determined that VWF is the predicted target of miR-424 and miR-15a, that FN1 is the predicted target of miR-424, miR-15a, and miR-199, and that A2M is the predicted target of miR-15a. 

The number of EVs isolated from the serum of the SC, CSC, and HC groups did not differ, a finding that agrees with other studies [2]. EV levels did not differ between the patients with stroke and the HCs. In contrast, other studies have found increased levels of circulating EVs in patients with stroke compared with HCs [25,26], which might reflect differences in the baseline characteristics of patients with stroke between studies and the different delays in collecting samples from symptom onset [25]. Previous studies have selected only patients with ischaemic stroke with NIHSS scores >5 [26], whereas our study included patients with NIHSS scores <5. Moreover, we did not find a significant correlation between EV levels at 24 h and final infarct volume at 1 week. Given that EV release as a consequence of stroke is not well understood and might be conditioned by various factors, it is important to analyse the profile of the EV cargo from HCs and patients with ischaemic stroke. 

### 4.1. Cargo of Circulating Extracellular Vesicles in Cortical-Subcortical Ischaemic Stroke

Our study detected the specific expression of C1QA in the EVs of the CSC group. C1QA is an essential part of innate immunity that acts by favouring the removal of infectious agents and apoptotic cells [27]. During cerebral ischaemia, C1QA modulates inflammation and participates in the clearance of damaged neurons and cellular debris. This anti-inflammatory environment limits neuronal stress and promotes neurite outgrowth and neurogenesis [28,29]. Our study revealed the role of C1QA inside EVs in reducing inflammation and enhancing brain plasticity in patients with CSC ischaemic stroke but not in those with SC involvement. Together with C1QA, A2M was one of the most abundant proteins in the CSC group. A2M is a large plasma glycoprotein that plays an important role in the interactions between several cytokines in the inflammation process [30]. Moreover, this protein is the predicted target for miR-15, and its levels were lower in the CSC group than in the SC group and in the HCs. The expression of miR-15 has also been related to the inflammation process as an enhancer of the induction of regulatory T cells [31]. C1QA, A2M, and miR-15 might therefore participate in the inflammation process in ischaemic stroke with CSC involvement. Moreover, C1QA has been shown to participate in another important stroke-related process as a risk factor for atherosclerosis. C1QA has been recognised as a mediator of atherosclerosis progression [32]. 

We identified several other important proteins whose levels inside circulating EVs were significantly higher in the CSC group than in the HCs and were related to atherosclerosis progression, such as FGC, FN1, and VWF [33,34]. These proteins have been shown to be responsible for platelet adhesion and aggregation during the formation of inflammatory processes. C1QA, FGG, FN1, and VWF could therefore be involved in atherogenesis progression in patients with CSC ischaemic stroke. These results agree with those of previous studies that determined that the EV cargo is implicated in atherosclerosis development, including immune responses, cell proliferation and migration, cell death, and vascular remodelling during disease progression [35]. This characteristic of the EV cargo in CSC ischaemic stroke might be due to the fact that most infarctions that affect the cortical and subcortical areas are large vessel strokes whose main aetiology is atherosclerosis. 

### 4.2. Cargo of Circulating Extracellular Vesicles in Subcortical Ischaemic Stroke

The present study detected the specific expression of ANXA2 in the EVs of the SC group in the pool of 26 patients, a result also validated in 10 isolated patients. ANXA is an annexin that belongs to the superfamily of calcium and phospholipid-binding proteins, a family that has diverse biological functions that include signal transduction during the inflammation process, potent regulation of the immune response, and anti-inflammatory effects. These results could indicate that ANXA2 has a differential expression in SC stroke but not in CSC stroke, and that it exerts an anti-inflammatory effect only in SC strokes. Previous studies have shown an increased ANXA2 expression after stroke in patients and in vitro studies [36,37]. Although these studies did not focus exclusively on SC strokes, their results agree with ours in that ANXA2 levels are elevated in patients with SC stroke compared with HCs. Moreover, previous studies have shown that ANXA2, together with the administration of tissue plasminogen activator (tPA), might contribute to the reduction of blood–brain barrier (BBB) disruption [38]. Consequently, ANXA2 diminishes the penetration of the blood component and frees tPA into the brain parenchyma. tPA in the parenchyma leads to microglia and astrocyte activation and associated detrimental pro-inflammatory cytokine release. All of the secondary neuroinflammatory responses can act as a feedback that results in further directly or indirectly disrupting BBB integrity [39]. Therefore, ANXA2 in the circulating EVs of patients with SC stroke can not only exert an anti-inflammatory effect but can also prevent BBB disruption in patients who undergo treatment with tPA. 

### 4.3. Cargo of Circulating Neural-Derived Extracellular Vesicles

Numerous studies have suggested that EVs are released into the blood from brain cells. The search for circulating brain-derived EVs could have the benefit of improving enriched fractions containing higher concentrations of key molecules. In the present study, we analysed the proteomics of EVs of neuronal origin (L1CAM^+^ EVs) and demonstrated that at least the specific proteins for CSC and SC stroke (C1QA, Casp14, and ANXA2) and the most abundant proteins, APOB, C3, FN1, VWF, and FGG, originate from neurons. In CSC and SC ischaemic stroke, different brain regions containing different cells and fibre components are affected. These different cells can present a distinct EV profile in blood. These circulating EVs can be considered markers of underlying brain damage and the repair process in both types of stroke. Further studies in animal models of stroke and in larger series of patients are needed to confirm the role of EVs and their content after ischaemic stroke.

### 4.4. Extracellular Vesicle Cargo Related to Recovery

Our study measured the ability of miRNA levels to predict an improvement in NIHSS scores from baseline. When assessing the outcomes of patients with CSC and SC stroke, the differences in baseline characteristics and frequency of administration of reperfusion therapies should be considered. We therefore chose neurological improvement over functional outcome as the main endpoint and included reperfusion therapies as a covariate. We found no relationship between any of the findings and functional outcomes according to the modified Rankin Scale (mRS) score, which could be due to the small sample size and the fact that most of the included patients achieved an mRS < 2. Surprisingly, we found an association between miR-100 and improved NIHSS scores. Thus, lower miR-100 levels in circulating EVs appeared to be related to improvement over the baseline deficit. Previous studies have demonstrated that miR-100 downregulation could be related to an increase in VWF and that miR-100 could be involved in both enhancing new vessel formation [40] and in vascular remodelling after stroke in mice [41], thereby promoting recovery in patients with CSC stroke. Moreover, the bioinformatic analysis of miR-424 and miR-15 binding sites also identified VWF as its predicted target. Previous studies have shown that miR-15a and miR-424 are also particularly powerful in promoting postischaemic vascular remodelling and angiogenesis after stroke [42,43] and in other brain diseases [40]. Thus, the miR-100, miR-424, miR-15a, and VWF inside the EVs of our study patients with CSC stroke could be promoting angiogenesis and new vessel formation after stroke. The study of miR-100 function in brain tissue from animal models of ischaemic stroke would represent a novel approach to the understanding of the role of this miRNA in stroke outcomes. 

### 4.5. Study Limitations

This study has certain limitations. First, infarct volume and the baseline vascular disease burden can differ between CSC and SC stroke and can influence the cargo of circulating EVs. In light of the heterogeneity of strokes with differing topography and the significant differences in infarct volume and baseline vascular disease burden, we analysed the correlation between levels of EVs at baseline and infarct volume at 7 days using FLAIR and DWI and did not find any correlation between them. Moreover, we analysed these differences in the vascular disease burden using the Fazekas score. We found no differences in caps or pencil thin lining, smooth halos, extending into the deep white matter, punctate foci, beginning confluence of foci, or large confluent areas of white matter lesions between the CSC and SC groups. We found differences only in silent brain infarcts. Based on these results, we consider that the infarct volume and vascular disease burden might not affect the differences in EVs in our study. 

Moreover, the stroke treatment might significantly affect the EV cargo [44]. Our study found differences in the number of patients who underwent intravenous thrombolysis and mechanical thrombectomy when comparing the CSC and SC groups. Ideally, only untreated patients should be included in the study to ensure that the differences in EV cargo are due to stroke topography and not to the treatments undergone. However, including only those patients who did not undergo therapy is very difficult, given that most patients arrive inside the window for interventions. Future studies in animal models should compare the EV cargo in a CSC and SC model of ischaemic stroke undergoing treatment and compare it with that of no treatment.

Lastly, the complexity of proteomic instrumentation introduces numerous potential sources of variability [45]. In our study, there was a low correlation in the detected protein profiles between the pooled samples and the individual samples within the same experimental group. In order to make the validation stronger, we employed various proteomic techniques. We used Orbitrab for pooling and sequential window acquisition of all theoretical fragment ion spectra mass spectrometry (SWATH-MS) for individual samples.

## 5. Conclusions

An integrated analysis of the proteome and transcriptome revealed differences in circulating EV proteins and microRNA profiles depending on ischaemic stroke topography. Proteins and microRNAs from patients with CSC ischaemic stroke might be markers of neurogenesis, neurite outgrowth, inflammation, and atherosclerosis. Moreover, proteins and microRNAs in EVs from patients with SC ischaemic stroke could be considered as markers of anti-inflammatory processes and of reductions in BBB disruption.

## Figures and Tables

**Figure 1 biomedicines-09-00786-f001:**
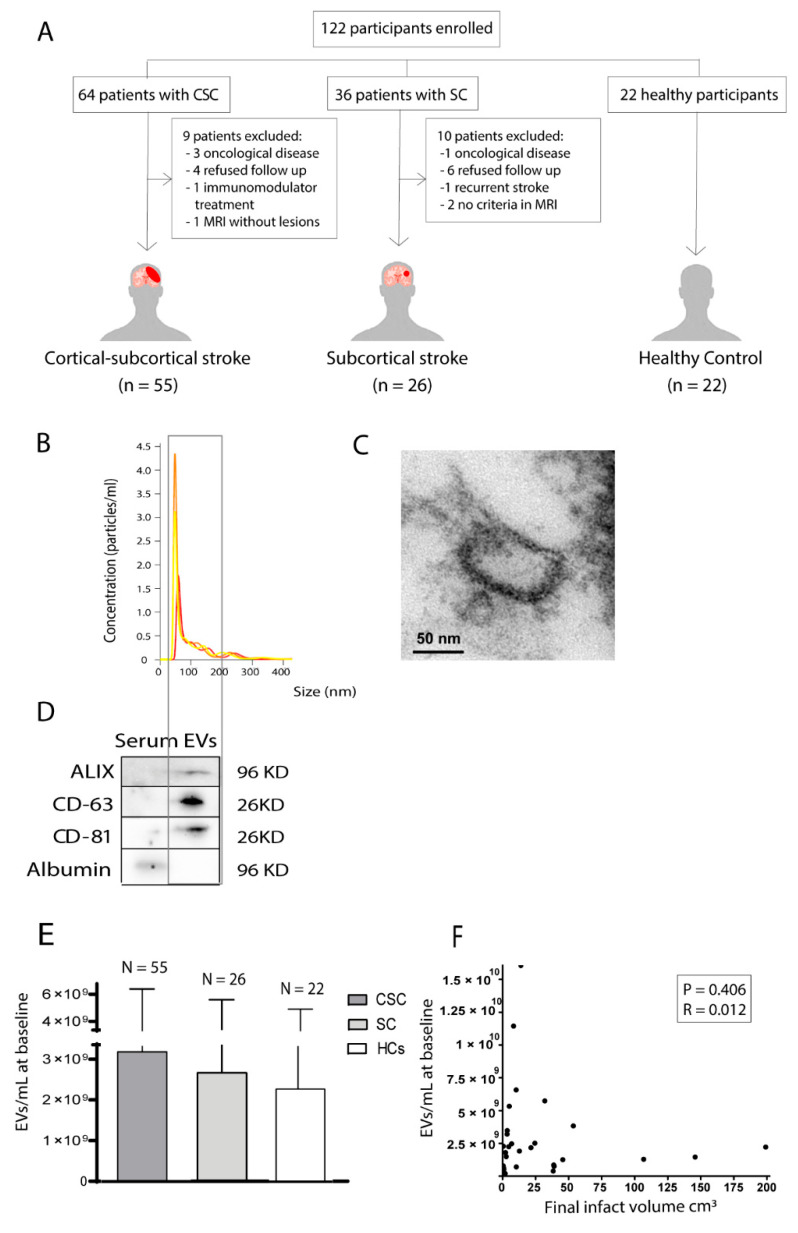
Participants, EV characterisation, and EV levels: (**A**) number of patients enrolled in the study; (**B**) characteristics of EVs. Size and concentration of the particles detected in the sample EVs isolated from HCs by NanoSight; (**C**) electron microscope image. EVs from HCs smaller than 100 nm were observed by electron microscope; (**D**) Western blot. Detection of EVs from HCs with specific markers (positive: Alix, CD63 and CD81; negative: Albumin) by Western blot. Negative control samples are serum. The gel image was cropped; (**E**) EV levels at 24 h showed no differences between the CSC, SC, and HC groups; (**F**) correlation between EV levels and total infarct volume. Data are expressed as mean ± SD. Abbreviations: CSC, cortical-subcortical stroke; EVs, extracellular vesicles; HCs, healthy controls; SC, subcortical stroke. The samples were run after only one freezing.

**Figure 2 biomedicines-09-00786-f002:**
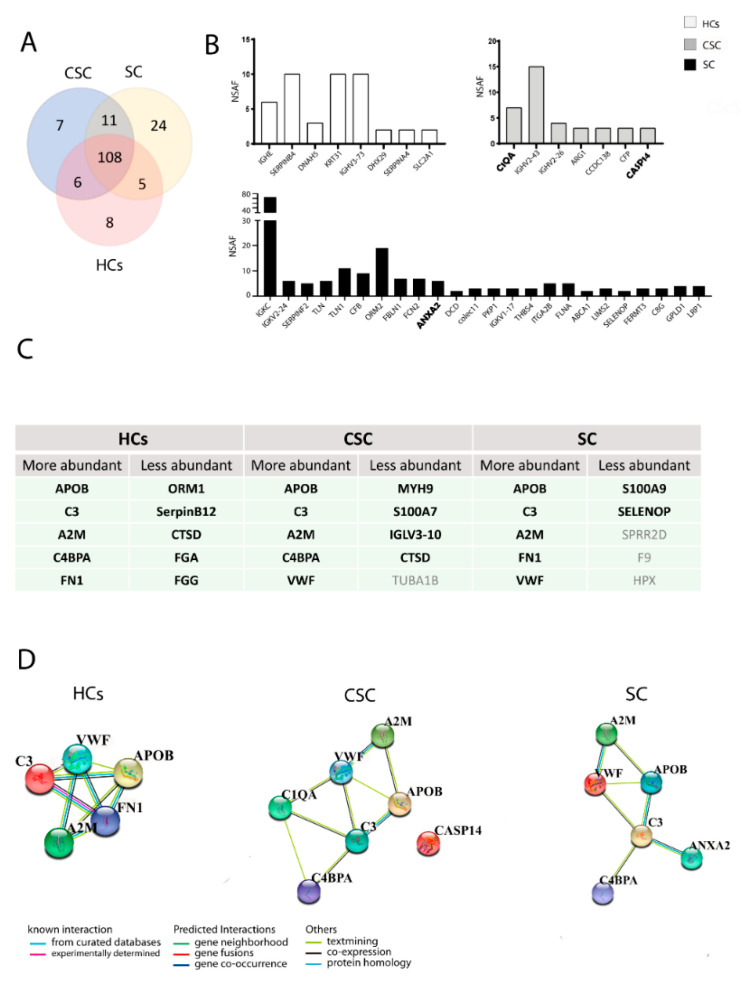
Significant proteins in the serum-derived EVs from healthy controls and patients with ischaemic stroke: (**A**) the distribution of detected proteins across the groups is represented by a Venn diagram; (**B**) the graphic represents the specific proteins found in each group. Proteins in bold have been validated in each group; (**C**) the table contains the more abundant and less abundant proteins in each group. Proteins in bold have been validated in each group; (**D**) interaction between the important proteins of each group (coloured nodes are query proteins and the first shell of interactors). Abbreviations: CSC, cortical-subcortical stroke; HCs, healthy controls; SC, subcortical stroke.

**Figure 3 biomedicines-09-00786-f003:**
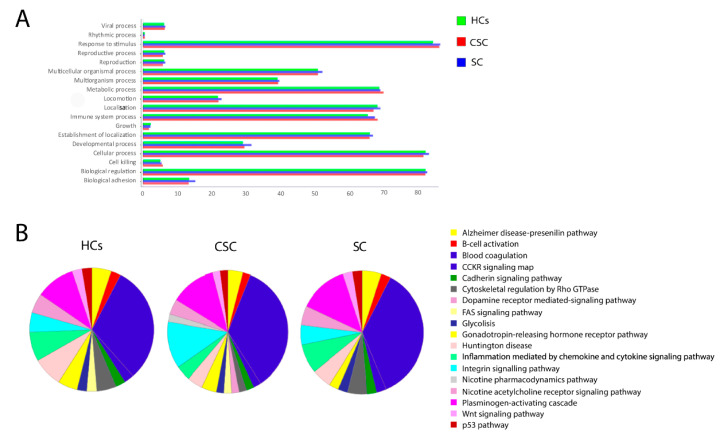
Gene ontology functional classification of serum-derived EV proteomes: (**A**) biological process; (**B**) pathways. Abbreviations: CSC, cortical-subcortical stroke; HCs, healthy controls; SC, subcortical stroke.

**Figure 4 biomedicines-09-00786-f004:**
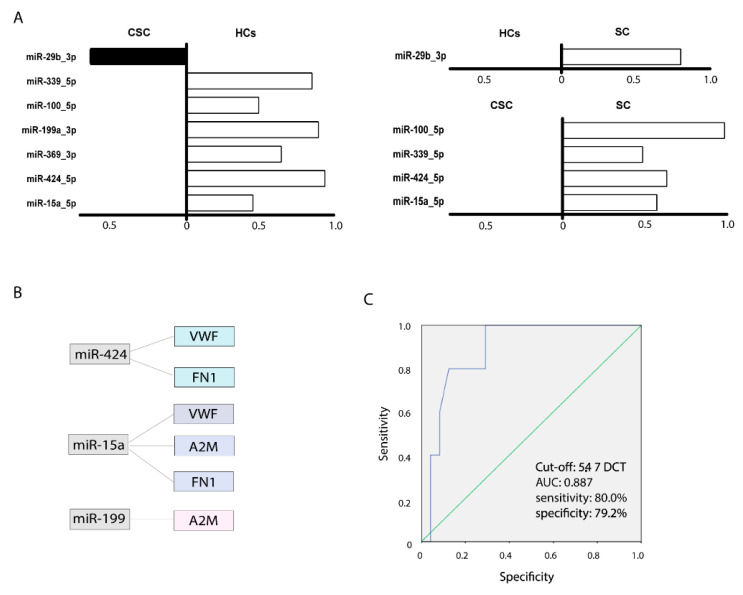
Significant miRNAs in serum-derived extracellular vesicles of healthy controls and patients with ischaemic stroke: (**A**) miRNA levels in patients with CSC stroke compared with HCs (left), HCs compared with patients with SC stroke (above right), and patients with CSC stroke compared with patients with SC (below right); (**B**) bioinformatics analysis of microRNAs and their predicted targets; (**C**) ROC curve representing the cut-off point corresponding to the maximum sensitivity and specificity values in order to identify the miRNA-100-5p levels to predict a >50% improvement from baseline NIHSS score at 3 months. Abbreviations: A2M, alpha-2-macroglobulin; AUC, area under the curve; CSC, cortical-subcortical stroke; FN1, fibronectin 1; HCs, healthy controls; SC, subcortical stroke; VWF, Von Willebrand factor.

**Table 1 biomedicines-09-00786-t001:** Demographic data, medical history, and stroke data at admission.

	CSC *n* = 55	SC *n* = 26	HC *n* = 22	*p*
Demographic Data				
Age, years [mean (SD)]	70.65 (15.39) *	61.65 (11.83)	61 (12.74) ^†^	**0.003**
Male, *n* (%)	24 (43.6)	18 (69.2)	7 (31.8)	**0.002**
Medical History				
Hypertension, *n* (%)	35 (63.6)	19 (73.1)	3 (13.63)	**0.001**
Diabetes mellitus, *n* (%)	10 (18.2)	5 (19.2)	2 (9.09)	0.53
Dyslipidaemia, *n* (%)	25 (45.5)	11 (42.3)	3 (13.63) ^†^	**0.001**
Smoker, *n* (%)	9 (16.4)	9 (34.6)	3 (13.63)	**0.001**
Coronary arterial disease, *n* (%)	6 (10.9%)	0 (0%)	0 (0%)	0.064
Atrial fibrillation, *n* (%)	15 (27.27%)	2 (7.69%)	0 (0%) ^†^	**0.004**
Infection, *n* (%)	6 (10.9%)	1 (3.84%)	0 (0%)	0.178
Charlson comorbidity index, (median; IQR)	1.00 (2.00)	1.00 (2.25)	0.00 (1.00) ^†^	**0.001**
Treatments				
Patients undergoing reperfusion therapy, *n* (%)	34 (61.81%)	5 (19.23%)	--	**<0.001**
IVT alone, *n* (%)	18 (34.54)	5 (19.2)	--	**0.001**
MT alone, *n* (%)	3 (11.53)	0 (0)	--	**0.002**
IVT and MT, *n* (%)	13 (32.73)	0 (0)	--	**0.008**
Recovery				
Baseline NIHSS (median; IQR)	9.00 (9.75)	3.00 (3.25)	--	**<0.001**
3 months NIHSS (median; IQR)	0.00 (1.00)	0.00 (1.00)	--	0.159
3 months mRS (median; IQR)	1.00 (2.00)	1.00 (2.00)	--	0.265
Percentage improvement from baseline NIHSS, mean (SD)	87.59% (22.85%)	78.65% (31.45%)	--	0.249
Patients who improved more than 50% baseline NIHSS, *n* (%)	37 (67.27%)	17 (56.38%)	--	0.090
Deaths, *n* (%)	1 (1.81%)	0 (0%)	--	0.323
Final Infarct Volume				
FLAIR, cm^3^ [mean (SD)]	30.95 (13.17)	3.15 (3.4)	--	**0.012**
DWI, cm^3^ [mean (SD)]	36.17 (50.23)	2.7 (2.6)	--	**0.008**
Baseline Vascular Disease Burden				
Fazekas PV score (median; IQR)	1(2)	1(2)	-	0.8
Fazekas DWM score (median; IQR)	1(2)	1(2)	-	0.759
Silent brain infarcts, *n* (%)	4 (7,1)	9 (36)	-	**0.002**

Abbreviations: CSC, cortical-subcortical stroke; cm^3^, cubic centimetres; DWI, diffusion-weighted imaging; FLAIR, fluid-attenuated inversion recovery; HC, healthy controls; IQR, interquartile range; IVT, intravenous thrombolysis; MT, mechanical thrombectomy; NIHSS, National Institutes of Health Stroke Scale score; SC, subcortical stroke; SD, standard deviation. Statistically significant values are in bold. * Mann–Whitney U test for continuous variables and Fisher’s exact test for categorical variables *p* < 0.016 to compare CSC and CS groups. ^†^ Mann–Whitney U test for continuous variables and Fisher’s exact test for categorical variables *p* < 0.016 compared with HCs. Comparison of previously published HC values with those measured in this study (CSC and SC values).

## Data Availability

The original data are available upon reasonable request.

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
