# Peer review of "Circulating Extracellular Vesicle Proteins and MicroRNA Profiles in Subcortical and Cortical-Subcortical Ischaemic Stroke"

_biomedicines, 2021, doi:10.3390/biomedicines9070786_

Round 1

Reviewer 1 Report

L1CAM in the peripheral extracellular vesicles can be of other tissue origins (e.g. lymphocytes, itestinal cells etc.) and cannot be conclusively indicate that these EV are of neuronal origin. 

The volume of stroke is an important parameter, especially involving different brain structures and had to be taken into account.

As authors stated in the study limitations, there was low correlation between individual's protein profiles and pooled samples, suggesting high heterogeneity of the samples, which suggests high variability of ALL the results. Inclusion of patients, which underwent intravenous thrombolysis and/or and mechanical thrombectomy makes the comparison between results in different groups very difficult to reconcile.

It seems that the data were not tested for the normality of distribution.

Reviewer 2 Report

The study by Laura Otero-Ortega et al compared the circulating extracellular vesicle proteins and microRNA profiles in subcortical and cortical-subcortical ischaemic stroke patients and proposed potential biomarkers for future clinical evaluations. 

several minor concerns need to be addressed.

  1. are the samples in FIgure 1 from HC,CSC or SC? this should be clarified in the figure legend.
  2. FIgure 4 B is not described in the legend.
  3. Result 2.3 and 2.8 should have figures.
